# Effects of One-Stage Full-Mouth Scaling and Root Planing with Azithromycin on Diabetes and Periodontal Disease: A Randomized Controlled Trial

**DOI:** 10.3390/antibiotics11091266

**Published:** 2022-09-18

**Authors:** Sho Komatsu, Shotaro Oshikiri, Takatoshi Nagano, Akihiro Yashima, Yuji Matsushima, Satoshi Shirakawa, Katsutoshi Komatsu, Akiko Mokubo, Kazuhiro Gomi

**Affiliations:** 1Department of Periodontology, Tsurumi University School of Dental Medicine, 2-1-3 Tsurumi, Tsurumi ku, Yokohama 230-8501, Japan; 2Department of Dental Hygiene, Tsurumi Junior College, 2-1-3 Tsurumi, Tsurumi ku, Yokohama 230-8501, Japan; 3Komatsu Clinic, 1989-66 Denbo, Fuji 417-0061, Japan; 4Mokubo Internal Medicine Clinic, 2-25 Kizukimotosumicho, Kawasaki 211-0021, Japan

**Keywords:** diabetes, full-mouth scaling and root planning, periodontal disease, inflammatory cytokines

## Abstract

Recent reports show that hemoglobin A1c (HbA1c) can be lowered by improving chronic inflammation in periodontal patients with diabetes mellitus and that full-mouth scaling and root planing (FM-SRP), in combination with azithromycin (AZM) treatment, can reduce early periodontal inflammation. However, the association of FM-SRP and AZM with periodontitis and HbA1c in patients with diabetes is largely unknown. This study investigated periodontitis and HbA1c in patients with diabetes after receiving FM-SRP and AZM to evaluate which clinical parameters most reflect the diabetic condition. Fifty-one periodontal patients with diabetes mellitus were included in this study. In total, 25 patients were assigned to the FM-SRP group in which patients were treated with FM-SRP in combination with AZM, and 26 patients were assigned to the control group in which only supragingival calculus removal was performed along with the provision of oral hygiene instructions. We evaluated periodontal parameters (probing pocket depth, periodontal inflamed surface area (PISA), bleeding on probing), and periodontal bacteria and biochemical parameters (HbA1c, high-sensitive C-reactive protein (hs-CRP), tumor necrosis factor-alpha (TNF-α), interleukin-6 (IL-6), and monocyte chemoattractant protein-1 (MCP-1)) at baseline (BL) and 1, 3, 6, and 9 months after treatment. Compared with BL values, the FM-SRP group showed improved clinical parameters, reduced periodontal pathogens, and significantly lower HbA1c. Inflammatory cytokines (hs-CRP, TNF-α, IL-6) were significantly reduced one month after treatment and remained low thereafter. MCP-1 did not change significantly during the experimental period. PISA showed a strong correlation with HbA1c, hs-CRP, and TNF-α. FM-SRP, in combination with AZM, produced clinical, microbiological, and HbA1c improvements in periodontal patients with previously diagnosed diabetes mellitus. Additionally, PISA was shown to be a useful index for assessing the diabetic status of patients with periodontal disease.

## 1. Introduction

The incidence of diabetes mellitus is on the rise worldwide. Complications resulting from diabetes include retinopathy, nephropathy, neuropathy, and periodontal disease [1,2]. The relationship between diabetes and periodontal disease has long been known, with type 2 diabetics reported to have a 2- to 3-fold higher incidence of periodontal disease [3] and more severe attachment loss and bone resorption than nondiabetics [4,5]. Additionally, it has been reported that the presence of severe periodontal disease may exacerbate HbA1c, increase insulin resistance, and influence the development and progression of cardiovascular disease or nephropathy [6,7,8]. The presence of periodontal disease is known to exacerbate diabetes and its complications. It has also been reported that comprehensive periodontal treatment, including antimicrobial therapy, scaling and root planing (SRP), extraction of nonpreservable teeth, and flap surgery for periodontal patients with diabetes, resulted in a statistically significant improvement in HbA1c of 0.5% after 3 months [9]. Additionally, when the treatment and control groups were further evaluated according to their high-sensitivity CRP (hs-CRP) values at entry, with patients categorized as either being in the high hs-CRP group (>0.05 mg/dL) or the low hs-CRP group value (<0.05 mg/dL), it can be concluded that periodontal treatment can significantly reduce HbA1c in the presence of inflammation as assessed by hs-CRP due to periodontitis [10]. Furthermore, systematic reviews and meta-analyses have shown that oral hygiene instruction, scaling, and root planing, as part of the periodontal treatment of patients with diabetes and periodontal disease, can improve HbA1c levels by a statistically significant 0.36–0.4% 3 to 4 months after treatment [11,12]. This is in addition to the regular oral care recommended for patients with diabetes mellitus. In contrast, a study found that the test group, which received oral hygiene instruction, SRP, and chlorhexidine rinsing, showed no improvement in HbA1c levels, although the condition of periodontal disease was significantly improved compared to the control group, which did not receive any treatment [13]. A 6-month follow-up after SRP showed a trend toward improvement in HbA1c, but no clearly significant difference was observed [14].

Conventional SRP is usually performed in several appointments in a quadrant-wise or sextant-wise manner, and it usually takes 1 to 3 months to complete the full-mouth treatment depending on the frequency of SRP. It was reported that periodontal pathogens, which were detected from the peri-implant sulcus of periodontally diseased patients, had the same genotype of pathogenic bacteria collected from the periodontal pockets of diseased natural teeth [15]. This report indicates that periodontopathic bacteria can translocate within the same oral cavity. When treatment is prolonged, periodontal-disease-related bacteria in untreated pockets may transmit to the treated sites, which are temporarily disinfected by mechanical debridement. It is possible that periodontal disease may relapse by reinfection. Therefore, to prevent bacterial reinfection, a combination of FM-SRP, which eliminates the infection at once by performing SRP on the whole mouth once, and antibiotics was performed. We selected azithromycin (2000 mg once a day: Zithromax SR, Pfizer, Tokyo, Japan) as the antibacterial agent. Azithromycin is the first of a subclass of macrolides called azalides, and it is more effective than earlier macrolides against a wide variety of organisms found in the oral cavity. It shows bacteriostatic activity against a wide variety of oral bacteria in vitro, has stronger antibiotic effects on Gram-negative microorganisms, including periodontopathic bacteria, compared to past macrolides, has a long half-life, and shows good tissue penetration. In addition, azithromycin is preferentially taken up by phagocytes, and therefore its level in infected tissues is higher than in non-infected sites [16,17,18,19]. It has been reported that full-mouth SRP (FM-SRP), in combination with azithromycin treatment in healthy subjects, improved periodontal disease symptoms such as inflammation (measured by bleeding on probing; BOP) more quickly and stabilized them over six months [20,21].

Therefore, this study consisted of an interventional study of FM-SRP with AZM in patients with type 2 diabetes and periodontal disease. We investigated the effect of FM-SRP with AZM on periodontal disease and diabetes in patients with diabetes and the relationship between periodontal parameters and diabetes.

## 2. Materials and Methods

### 2.1. Subjects

This clinical study was conducted as a single-blind randomized controlled trial in two private practices from 2019 to 2021. The subjects were recruited among patients with type 2 diabetes and periodontal disease from diabetic medical institutions. Fifty-one patients with periodontitis (Stage 2 or 3, Grade B) were registered. This study was conducted under the approval of the Ethics Committee of Tsurumi University School of Dental Medicine (Approval No. 1620) and registered in the clinical trial database (UMIN000035472, https://www.umin.ac.jp (accessed on 11 December 2021)). This study was then conducted in accordance with the ethical standards of the 1975 Declaration of Helsinki, as revised in 2013. All subjects gave their written consent with full informed consent.

The inclusion criteria were as follows: (a) patients with HbA1c >6.5%, (b) patients aged >35 years, (c) patients with at least 15 teeth, and (d) patients who were nonsmokers. The exclusion criteria were as follows: (a) patients who had serious systemic diseases other than diabetes, (b) patients who had received periodontal treatment within the previous six months, (c) patients who had received systemic and local antibiotics within the past three months, and (d) patients with a history of hypersensitivity to macrolide antibiotics containing AZM.

### 2.2. Randomization and Allocation Concealment

From a total of 124 applicants, 51 subjects were selected according to the selection criteria. Subjects were randomly assigned to the control group (*n* = 26) and the FM-SRP group (*n* = 25) using the envelope method. Randomization was performed using sealed envelopes containing the name of one of the two groups. Group assignment was revealed to the dentist in charge immediately before SRP. However, the dentist in charge of the examination was blinded and not informed of the group assignment until the end of the study. It is necessary to use FM-SRP alone as a control for FM-SRP + AZM, but the ethics committee has stated that it is ethically problematic to perform FM-SRP alone without antibiotics in diabetic patients who are susceptible to infections. Therefore, the control group only had scaling performed with a low risk of infection.

### 2.3. Clinical Protocol

Figure 1 shows the study design flowchart. The subjects underwent periodontal examination after the collection of bacterial samples and peripheral blood from the median cubital vein at BL. Three days before FM-SRP, the FM-SRP group was given AZM (2000 mg once a day: Zithromax SR, Pfizer, Japan). All scaling (SC), FM-SRP, and professional mechanical tooth-cleaning (PMTC) were performed by one dentist. FM-SRP was performed under local anesthesia ((2–3) 1.8 mL 2% lidocaine, 1:80,000 epinephrine) using Gracey curettes (YDM Corporation, Tokyo, Japan) and an ultrasonic scaler (P-MAX, Satelec Ltd., Mérignac, France) during a single visit. The control group received oral cleaning instructions, SC using an ultrasonic scaler, and PMTC from the same dentist who treated the FM-SRP group. All subjects underwent clinical and bacteriological examinations and peripheral blood sampling for biochemical tests 1, 3, 6, and 9 months after treatment. A different dentist performed the laboratory tests, bacterial tests, and blood sampling from the dentists who had performed SC, SRP, and PMTC. This dentist was not informed of the group allocations. Oral hygiene instruction and SC were provided during the visit (Figure 1).

### 2.4. Clinical Examination

Probing depth (PD) and BOP were recorded at six sites (proximal buccal, buccal, distal buccal, proximal lingual, lingual, and distal lingual) per tooth. Additionally, the value of periodontal inflamed surface area (PISA) from PD and BOP [22] was determined and recorded. PISA is a quantification of BOP-positive sites in the epithelial area in the periodontal pocket, indicated in square millimeters (mm^2^). PISA was calculated by entering the PPD and BOP into the PISA calculation spreadsheet [22] published on the Internet website [23] according to the report by Nesse et al. [24].

### 2.5. Bacteriological Examination

Bacterial samples were collected using sterile paper points (#40, Zipperer Absorbent Paper Points, VDW GmbH, Munich, Germany) from the two deepest periodontal pockets for 30 s. The paper points were placed in 1 mL of sterilized distilled water and stored at −80 °C. Bacteriological samples were analyzed by BML Corporation (Tokyo, Japan). The quantitative analysis of the number of *Porphyromonas*
*gingivalis*, *Tannerella*
*forsythia*, *Treponema*
*denticola*, *Prevotella intermedia*, *Aggregatibactter actionmycetemcomitans*, and total bacteria was performed using a modified Invader PLUS assay. The detailed measurement method has been described in previous reports [25,26].

### 2.6. Biochemical Examinations

Peripheral blood samples were obtained by venipuncture. HbA1c, and the inflammatory markers hs-CRP, TNF-α, IL-6, and MCP-1, were analyzed in laboratories (Hoken Kagaku, Inc. Yokohama, Japan). HbA1c levels were determined by an enzymatic method using MetaboLeadHbA1c (Minaris Medical Co., Ltd., Tokyo, Japan). The hs-CRP levels were determined using an N-assay LA CRP-U (Nittobo Medical Co., Ltd., Tokyo, Japan). TNF-α levels were determined by ELISA, using a Quantikine HS Human TNF-1α Immunoassay (R&D SYSTEMS, Inc., Minneapolis, MN, USA). IL-6 was measured by the chemiluminescent enzyme immunoassay (CLEIA) method using a human IL-6 measuring cartridge (FUJIREBIO Co., Ltd., Tokyo, Japan). MCP-1 levels were measured via an enzyme immunoassay (EIA) (Quantikine Human MCP-1 Immunoassay; Funakoshi Co, Ltd., Tokyo, Japan).

### 2.7. Statistical Analysis

All collected data were sent to Tsurumi University in a password-protected Excel document, with patients’ personal information removed. All data were then combined and analyzed on a subject-by-subject basis. Results are presented as mean ± SD. Student’s *t*-test and Fisher’s exact test were used for comparing age and gender at BL between the FM-SRP and control groups. Additionally, as a result of investigating the distribution state of the data using the Shapiro–Wilk test, clinical parameters (PD, BOP, and PISA), periodontal pathogen count, HbA1c, body mass index (BMI), and inflammatory markers (hs-CRP, TNF-α, IL-6, and MCP-1) were found not to show a normal distribution, so the analysis was considered a nonparametric test. The Mann–Whitney *U* test was used to compare the FM-SRP and control groups for these factors. Nine-month changes in each parameter were analyzed within each group using the Friedman test. Multiple comparisons were then tested for statistical significance with the Wilcoxon signed-rank test with Bonferroni correction. The *p* value was adjusted by adjusting Bonferroni for all 10 combinations of 5 time points, and the significance level was set at *p* < 0.005 of 0.05/10. Correlations between clinical parameters (PD, BOP, and PISA), HbA1c, BMI, and inflammatory markers (hs-CRP, TNF-α, IL-6, and MCP-1) were evaluated using Spearman’s rank correlation coefficient. Bonferroni correction was used to adjust the *p* value for multiple comparisons to assess the statistical significance of the correlation between these factors. Since the number of tests was 28, the significance level of the correlation was set to *p* < 0.0017 (0.05/28) based on the Bonferroni correction. All analyses were performed using the appropriate software (JMP14.0.0, SAS Institute Inc., Japan). A sample size calculation suggested that at least 6 patients were needed to demonstrate a 0.5% decrease in HbA1c after FM-SRP (95% power, α = 0.01, a standard deviation of 0.18 (JMP14.0.0)) [9].

## 3. Results

From a total of 124 applicants, 51 subjects were selected according to the selection criteria and randomly divided into 2 groups: the FM-SRP group (25 subjects) and the control group (26 subjects). During the experimental period, one subject from the FM-SRP group and four from the control group were excluded from the analysis because they never visited the clinic after treatment. In total, 24 subjects in the FM-SRP group and 22 in the control group completed the entire study protocol. No adverse reactions, including diarrhea, were observed during the experimental period. There were no changes by physicians in diet, exercise, or pharmacological treatment of the patients’ diabetes mellitus. Patient age averaged 66.7 ± 13.9 (36–88) in the FM-SRP group and 63.5 ± 12.2 (36–82) in the control group. HbA1c averaged 7.4 ± 0.4 (6.5–8.2) % and 7.3 ± 0.6 (6.7–9.2) % in the FM-SRP and control groups, respectively. BMI averaged 26.3 ± 4.8 (20.7–40.6) kg/m^2^ and 24.8 ± 5.3 (16.3–40.9) kg/m^2^ in the FM-SRP and control groups, respectively. No significant differences were observed in BL for any clinical parameters (PPD, BOP, and PISA), BMI, HbA1c, bacterial counts, or inflammatory markers (hs-CRP, TNF-α, IL-6, MCP-1) between the FM-SRP group and the control group (Table 1).

### 3.1. Clinical Parameters

Among all the clinical parameters measured (PPD, BOP, and PISA), the control group showed no significant changes relative to the BL and any time point. In contrast, the FM-SRP group showed significant improvement in the clinical parameters compared to BL at 1, 3, 6, and 9 months (*p* < 0.001). Additionally, the FM-SRP group showed significant improvement over the control group at all time points (*p* < 0.001) (Table 2).

### 3.2. Bacteriological Examination

Among the FM-SRP group, *P. intermedia*, *P. gingivalis*, *T. forsythia*, and *T. denticola* levels significantly decreased after treatment. In contrast, there was no change in the number of bacteria observed in the control group. The FM-SRP group showed significant reductions in all bacteria except *A. actinomycetemcomitans* at each time point compared to the control group, and these reductions were maintained for the nine-month duration of the study. It is noteworthy that *A. actinomycetemcomitans* appeared in only a few patients, and no significant differences were observed within or between groups (Table 3).

### 3.3. Biochemical Examinations

#### 3.3.1. HbA1c

The HbA1c levels observed in the control group at BL were 7.3 ± 0.6%, and 7.4 ± 0.4% in the FM-SRP group, which was not significantly different, but the FM-SRP group showed a significant decrease in HbA1c levels compared with BL, with 7.3 ± 0.4%, 7.2 ± 0.4%, 7.1 ± 0.5, and 7.1 ± 0.5%, at 1, 3, 6, and 9 months, respectively. Conversely, the control group showed no change in HbA1c levels compared with BL. Six months after treatment, a significant difference was observed between the control and FM-SRP groups (*p* < 0.05) (Table 4).

#### 3.3.2. hs-CRP

The mean hs-CRP levels were 0.08 ± 0.07 mg/dL in the control group and 0.19 ± 0.32 mg/dL in the FM-SRP group at BL, with no significant difference between the two. The control group showed no change in hs-CRP at any time point while the FM-SRP group showed a significant decrease at 1, 6, and 9 months compared to BL (*p* < 0.005). The FM-SRP group showed a significant difference from the control group at one month (*p* < 0.05) (Table 4).

#### 3.3.3. TNF-α

The mean TNF-α levels were 0.9 ± 0.3 pg/mL in the control group and 1.1 ± 0.5 pg/mL in the FM-SRP group at BL, with no significant difference between the two. The control group showed no change in TNF-α at all time points while the FM-SRP group showed a significant decrease at 1 and 3 months compared to BL (*p* < 0.005). The FM-SRP group showed a significant difference from the control group at one month (*p* < 0.05) (Table 4).

#### 3.3.4. IL-6

The mean IL-6 level in the control group at BL was 2.2 ± 0.6 pg/mL, and 2.3 ± 0.6 pg/mL in the FM-SRP group, with no significant difference between the two. The control group showed no change in IL-6 at any time point while the FM-SRP group showed a significant decrease at one month compared to BL (*p* < 0.005). Additionally, there was a significant difference in the IL-6 levels between the control and FM-SRP groups (*p* < 0.05) (Table 4).

#### 3.3.5. MCP-1

The mean MCP-1 in the control group at BL was 124.9 ± 50.7 pg/mL, and 120 ± 45.7 pg/mL in the FM-SRP group, with no significant difference between the two. Neither the control nor the FM-SRP group showed significant changes during the study period (Table 4).

### 3.4. BMI

At BL, the mean BMI was 24.8 ± 5.3 kg/m2 in the control group and 26.3 ± 4.8 ka/m^2^ in the FM-SRP group, with no significant difference observed. During this study period, no significant changes were observed within or between the FM-SRP and control groups (Table 4).

### 3.5. Correlation among HbA1c, Inflammatory Cytokines, and Clinical Parameters

To evaluate the overall relationships among HbA1c, clinical parameters, and inflammatory markers at BL, the correlations among these factors are presented in Table 5 for both study groups combined. Correlations of *p* < 0.001 (0.05/28 tests) were considered to be statistically significant based on the Bonferroni adjustment for multiple comparisons. HbA1c showed a significant correlation with hs-CRP (*r* = 0.5683, *p* < 0.0001) and TNF-α (*r* = 0.4823, *p* = 0.0007), and with PPD (*r* = 0.5629, *p* < 0.0001), BOP (*r* = 0.5061, *p* = 0.0003), and PISA (*r* = 0.5963, *p* < 0.0001). PISA showed significant correlations not only with HbA1c but also with hs-CRP (*r* = 0.4738, *p* = 0.0009), TNF-α (*r* = 0.4763, *p* = 0.0008), PPD (*r* = 0.8242, *p* < 0.0001), and BOP (*r* = 0.9353, *p* < 0.0001) (Table 5).

### 3.6. Correlation with Nine-Month Changes in the FM-SRP Group and the Control Group

A correlation with HbA1c was observed for hs-CRP (*p* < 0.002, 0.5995) and TNF-α (*p* < 0.0011, 0.6265) in the FM-SRP group, but no correlation was observed in the control group (Table 6 and Table 7).

## 4. Discussion

Patients with type 2 diabetes are known to have a higher incidence of periodontal disease, and poor glycemic control is associated with more severe periodontal disease [27,28,29,30]. Various reports have investigated whether periodontal treatment improves the symptoms of diabetes. Since periodontal disease is caused by bacterial infection, it is considered that further improvements could be observed by applying antibacterial agents to periodontal treatment. However, there have been studies in which basic periodontal therapy with SRP and doxycycline (100 mg/day for 14 days) in patients with type 2 diabetes showed no significant differences in clinical parameters after 3 to 6 months [31,32]. Additionally, a study of patients with type 2 diabetes who received FM-SRP alone compared with FM-SRP combined with oral amoxicillin administered within 24 h reported no significant difference in the improvement of clinical parameters at 3 months between the two groups [33]. On the other hand, another study reported that SRP, combined with a combined oral administration of metronidazole and amoxicillin in patients with type 2 diabetes and extensive chronic periodontitis, showed significant improvement in clinical parameters compared to SRP treatment alone [34].

In the present study, the FM-SRP group showed significant improvements in all clinical parameters (PPD, BOP, and PISA) after treatment, which was maintained for nine months. In particular, BOP and PISA greatly improved, and these findings were consistent with those of a study in which FM-SRP in combination with AZM was given to healthy subjects [20]. This suggests that FM-SRP with AZM is also effective in improving periodontal disease in patients with diabetes affected by periodontal disease and that inflammation improves more quickly. The reason for this difference in study results using SRP with antimicrobial agents may be due to differences in the method used to perform SRP, the agents used, and the pharmacokinetics of those agents. AZM is taken up by phagocytes, accumulates specifically at inflammatory sites, and has a long duration of action [16]. This may result in a greater local effect on inflammation than blood concentration-dependent antimicrobials. AZM may be more effective than other antimicrobials due to this factor, but further study is needed to confirm this.

Many previous studies have reported significantly higher detection rates of *P. gingi-valis* in the periodontal pockets of patients with diabetes mellitus and periodontal disease [35,36,37,38]. Sbordone et al. [39] also reported significantly higher frequencies of *P. gingivalis* and *Capnocytophaga* spp. and lower frequencies of *P. intermedia* and *A. actinomycetemcomitans.* In the present study, *P. intermedia*, *P. gingivalis*, *T. forsythia*, and *T. denticola* were detected in patients quite often while *A. actinomycetemcomitans* was not evaluated due to its infrequent incidence at BL. As described, there are various reports on the frequency of bacteria in patients with diabetes, and it is difficult to draw a general conclusion because of the influence of race, lifestyle, and other factors.

In this study, the number of periodontopathic bacteria in the FM-SRP group was significantly lower for *P. intermedia*, *P. gingivalis*, *T. forsythia*, and *T. denticola* at all time points compared to BL. Additionally, compared with the control group, the periodontopathic bacteria in the FM-SRP group were significantly lower than that of the control group at all time points. This may indicate that in combination with AZM, FM-SRP reduces periodontopathic bacteria and improves periodontal pocket flora, thereby improving periodontal tissue inflammation and maintaining clinical stability.

Although there have been reports [40] that inflammation or metabolic parameters in patients with diabetes are affected by BMI, we believe that the influence of BMI can be excluded in this study because there was no significant difference between the FM-SRP and control group.

In a study of SRP in periodontal patients with diabetes, it was reported that there was no change in HbA1c and serum biomarkers (hs-CRP, TNF-α, IL-6) during the 6-month follow-up period, regardless of whether SRP was performed and the severity of periodontal disease [41]. In contrast, it was reported that the experimental group that underwent intensive periodontal treatment (periodontal surgery or subgingival calculus removal) showed significant improvement in clinical parameters and inflammatory cytokines (CRP, TNF-α), and HbA1c at 12 months compared to the control group that underwent supragingival calculus removal and polishing [42]. In the present study, the FM-SRP group showed a significant improvement in clinical parameters, a significant decrease in periodontal pathogen counts, and a significant decrease in HbA1c compared to BL from 1 to 9 months after treatment. These results show that FM-SRP, although a nonsurgical periodontal treatment, demonstrated clinical and biochemical improvements similar to those seen in the previously described surgical periodontal treatment intervention. Additionally, HbA1c decreased by approximately 0.2–0.3%, consistent with previous studies [43]. Conversely, no significant change in HbA1c was observed in the control group, in which BOP, PISA, and inflammatory cytokines were not decreased, suggesting that inflammatory factors strongly influence HbA1c. Previous studies have shown a strong correlation between TNF-α and HbA1c [44,45]. In the present study, hs-CRP and IL-6 were significantly decreased one month after treatment. TNF-α was also significantly decreased at one and three months after treatment. Inflammatory cytokines remained low after three months, although this was not significantly different, suggesting that the significant decrease in HbA1c was maintained over nine months. Furthermore, since there is no change in the expression of MCP-1 that reflects the inflammatory state in adipose tissue [46], it is considered that the change in HbA1c was a result of the improvement of the inflammation in the periodontal tissue.

Examining the relationship between BL in the combined FM-SRP and control groups and the amount of change after nine months, the biomarkers that showed correlation with HbA1c were hs-CRP and TNF-α while the clinical parameters PPD, BOP, and PISA all showed correlation. In addition, PISA showed a strong correlation with HbA1c, hs-CRP, and TNF-α, suggesting that PISA may be an effective index for evaluating periodontal disease and diabetic status in patients with diabetes and periodontal disease. However, this study was limited by the small number of patients enrolled, and further studies with larger sample sizes are needed.

In conclusion, FM-SRP combined with AZM in periodontal patients with diabetes mellitus could quickly improve clinical parameters, periodontal pathogenic bacteria counts, and HbA1c in a short time. Additionally, PISA was identified as a potentially useful index for evaluating diabetes status in patients with diabetes and periodontal disease because of its strong correlation with HbA1c, hs-CRP, and TNF-α.

## 5. Conclusions

FM-SRP, in combination with AZM, produced clinical, microbiological, and HbA1c improvements in periodontal patients with previously diagnosed diabetes mellitus. Additionally, PISA was shown to be a useful index for assessing the diabetic status of patients with periodontal disease.

## Figures and Tables

**Figure 1 antibiotics-11-01266-f001:**
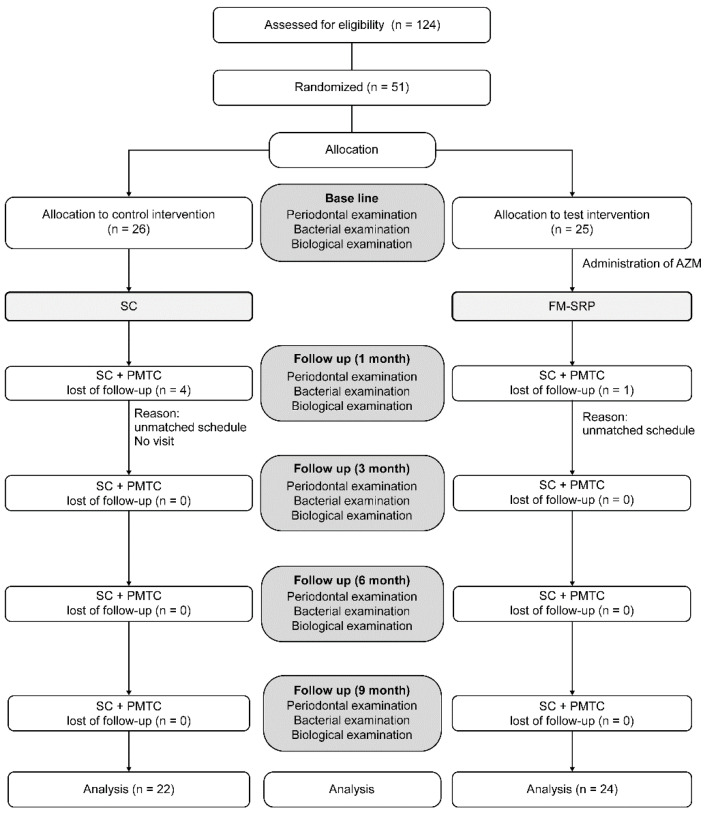
Study design flowchart.

**Table 1 antibiotics-11-01266-t001:** BL values.

	FM-SRP (*n* = 24)	Control (*n* = 22)	*p* Value
Age	66.7 ± 13.9 (36–88)	63.5 ± 12.2 (36–82)	0.4121
Sex (male/female)	24 (9/15)	22 (8/14)	0.9364
HbA1c (%)	7.4 ± 0.4 (6.5–8.2)	7.3 ± 0.6 (6.7–9.2)	0.067
BMI (kg/m^2^)	26.3 ± 4.8 (20.7–40.6)	24.8 ± 5.3 (16.3–40.9)	0.3673
PPD (mm)	3.6 ± 0.5	3.5 ± 0.3	0.6189
BOP (%)	48.2 ± 15.4	44.6 ± 10.8	0.448
PISA (mm^2^)	884.9 ± 395.9	884.4 ± 235.1	0.6207
hs-CRP (mg/dL)	0.19 ± 0.32	0.08 ± 0.07	0.2222
TNF-α (pg/mL)	1.1 ± 0.5	0.9 ± 0.3	0.1183
IL-6 (pg/mL)	2.3 ± 0.6	2.2 ± 0.6	0.6199
MCP-1 (pg/mL)	120.0 ± 45.7	124.9 ± 50.7	0.5674

Values are expressed as mean ± SD. Student’s *t*-test and Fisher’s exact test were used to compare age and gender at BL between the FM-SRP and control groups. The Mann–Whitney’s *U* test was used to compare clinical parameters (PD, BOP, and PISA), periodontal pathogen count, HbA1c, BMI, and inflammatory markers (hs-CRP, TNF-α, IL-6, and MCP-1) between groups. The significance level between groups was set at *: *p* < 0.05.

**Table 2 antibiotics-11-01266-t002:** Clinical parameters.

	Time	FM-SRP	*p* Value (vs. BL)	Control	*p* Value (vs. BL)	*p* Value (FM-SRP vs. Control)
PPD (mm)	BL	3.6 ± 0.5	-	3.5 ± 0.3	-	0.6189
	1 M	2.5 ± 0.3	<0.0001 *	3.3 ± 0.3	0.0097	<0.0001 *
	3 M	2.3 ± 0.2	<0.0001 *	3.4 ± 0.3	0.0321	<0.0001 *
	6 M	2.5 ± 0.2	<0.0001 *	3.3 ± 0.3	0.0206	<0.0001 *
	9 M	2.5 ± 0.2	<0.0001 *	3.4 ± 0.3	0.054	<0.0001 *
BOP (%)	BL	48.2 ± 15.4	-	44.6 ± 10.8	-	0.448
	1 M	9.5 ± 7.9	<0.0001 *	43.4 ± 9.2	0.0826	<0.0001 *
	3 M	6.9 ± 5.6	<0.0001 *	43.2 ± 9.3	0.0289	<0.0001 *
	6 M	6.4 ± 5.5	<0.0001 *	43.8 ± 9.8	0.2377	<0.0001 *
	9 M	6.1 ± 4.7	<0.0001 *	43.5 ± 9.5	0.4196	<0.0001 *
PISA (mm^2^)	BL	884.9 ± 395.9	-	884.4 ± 235.1	-	0.6207
	1 M	100.5 ± 92.5	<0.0001 *	821.4 ± 227.6	0.0074	<0.0001 *
	3 M	94.1 ± 118.0	<0.0001 *	830.3 ± 228.9	0.0132	<0.0001 *
	6 M	98.7 ± 131.8	<0.0001 *	828.3 ± 206.6	0.0057	<0.0001 *
	9 M	66.9 ± 53.6	<0.0001 *	833.9 ± 197.6	0.0703	<0.0001 *

Values are expressed as mean ± SD. Comparison of clinical parameters (PPD, BOP, PISA) values between BL and each time point was performed using the Wilcoxon’s signed-rank test and further adjusted for multiple comparisons with a significance level of *: *p* < 0.005. Comparisons between groups were conducted using the Mann–Whitney *U* test with a significance level of *: *p* < 0.05.

**Table 3 antibiotics-11-01266-t003:** Periodontal pathogen counts.

	Time	FM-SRP	*p* Value (vs. BL)	Control	*p* Value (vs. BL)	*p* Value (FM-SRP vs. Control)
*P. gingivalis*	BL	2.1 ± 1.1	-	1.6 ± 1.0	-	0.1799
(log10)	1 M	1.0 ± 0	<0.0001 *	1.8 ± 2.0	0.2374	0.0014 *
	3 M	1.1 ± 0.2	<0.0001 *	1.6 ± 0.9	0.4767	0.0308 *
	6 M	1.2 ± 0.5	0.0002 *	1.7 ± 1.0	0.9463	0.0122 *
	9 M	1.1 ± 0.4	0.0002 *	1.9 ± 1.1	0.3565	0.0167 *
*T. denticola*	BL	2.1 ± 0.9	-	1.9 ± 0.8	-	0.315
(log10)	1 M	1.1 ± 0.3	<0.0001 *	1.7 ± 0.9	0.316	0.0015 *
	3 M	1.2 ± 0.4	<0.0001 *	1.8 ± 1.0	0.6392	0.0024 *
	6 M	1.1 ± 0.3	<0.0001 *	1.8 ± 0.8	0.4297	<0.0001 *
	9 M	1.1 ± 0.4	0.0002 *	1.8 ± 0.9	0.5789	0.0026 *
*T. forsythia*	BL	1.8 ± 0.7	-	1.6 ± 0.7	-	0.2765
(log10)	1 M	1.1 ± 0.2	<0.0001 *	1.6 ± 0.7	0.8748	0.0009 *
	3 M	1.2 ± 0.4	0.0023 *	1.7 ± 0.6	0.1812	0.0021 *
	6 M	1.2 ± 0.5	0.0027 *	1.9 ± 0.7	0.1776	0.0003 *
	9 M	1.3 ± 0.5	0.0032 *	1.9 ± 0.7	0.1373	0.0009 *
*P. intermedia*	BL	1.4 ± 0.6	-	1.3 ± 0.4	-	0.5936
(log10)	1 M	1.1 ± 0.3	0.0013 *	1.7 ± 2.0	0.9195	0.0036 *
	3 M	1.1 ± 0.2	0.0013 *	1.5 ± 1.0	0.2264	0.0089 *
	6 M	1.1 ± 0.2	0.0013 *	1.3 ± 0.4	0.6485	0.0236 *
	9 M	1.1 ± 0.2	0.0013 *	1.3 ± 0.4	0.4151	0.0132 *
*A. actinomycetemcomitans*	BL	1.0 ± 0.10	-	1.0 ± 0.12	-	0.9104
(log10)	1 M	1.0 ± 0	0.1618	1.5 ± 1.9	0.54	0.1353
	3 M	1.0 ± 0	0.1618	1.1 ± 0.4	0.1622	0.1353
	6 M	1.0 ± 0	0.1618	1.0 ± 0.02	0.1622	0.2963
	9 M	1.0 ± 0.2	0.9768	1.0 ± 0.09	0.1622	0.9752

Values are expressed as mean ± SD. Comparisons between BL and time points for each bacterial count were made by the Wilcoxon signed-rank test with adjustment for multiple comparisons at a significance level of **p* < 0.005. Comparisons between groups were conducted using the Mann–Whitney *U* test with a significance level of *: *p* < 0.05.

**Table 4 antibiotics-11-01266-t004:** HbA1c and inflammatory markers.

	Time	FM-SRP	*p* Value (vs. BL)	Control	*p* Value (vs. BL)	*p* Value (FM-SRP vs. Control)
HbA1c (%)	BL	7.4 ± 0.4	-	7.3 ± 0.6	-	0.067
	1 M	7.3 ± 0.4	0.0017 *	7.4 ± 0.7	0.2147	0.9121
	3 M	7.2 ± 0.4	0.0005 *	7.5 ± 0.9	0.0363	0.275
	6 M	7.1 ± 0.5	<0.0001 *	7.6 ± 1.0	0.0322	0.0291 *
	9 M	7.1 ± 0.5	0.0001 *	7.4 ± 0.7	0.0808	0.0552
hs-CRP	BL	0.19 ± 0.32	-	0.08 ± 0.07	-	0.2222
(mg/dL)	1 M	0.06 ± 0.06	<0.0001 *	0.10 ± 0.11	0.1836	0.0441 *
	3 M	0.09 ± 0.09	0.0101	0.07 ± 0.09	0.3668	0.5236
	6 M	0.08 ± 0.07	0.0030 *	0.07 ± 0.10	0.2583	0.6129
	9 M	0.07 ± 0.08	<0.0001 *	0.12 ± 0.26	0.3668	0.7414
TNF-α	BL	1.1 ± 0.5	-	0.9 ± 0.3	-	0.1183
(pg/mL)	1 M	0.76 ± 0.2	0.0002 *	0.9 ± 0.3	1	0.0453 *
	3 M	0.8 ± 0.3	0.0019 *	0.9 ± 0.3	0.2314	0.7001
	6 M	0.9 ± 0.2	0.0396	0.9 ± 0.3	0.8506	0.8258
	9 M	0.9 ± 0.2	0.0111	0.9 ± 0.3	0.3493	0.7833
IL-6 (pg/mL)	BL	2.3 ± 0.6	-	2.2 ± 0.6	-	0.6199
	1 M	1.8 ± 0.7	0.0045 *	2.3 ± 0.9	0.826	0.0185 *
	3 M	2.1 ± 0.6	0.033	2.1 ± 0.6	0.3578	0.7493
	6 M	2.2 ± 0.8	1	2.4 ± 0.6	0.2934	0.3633
	9 M	2.1 ± 0.6	0.4928	2.1 ± 0.6	0.9374	0.5814
MCP-1	BL	120.0 ± 45.7	-	124.9 ± 50.7	-	0.5674
(pg/mL)	1 M	118.2 ± 36.1	0.1527	120.4 ± 51.9	0.4779	0.373
	3 M	113.9 ± 36.4	0.879	130.3 ± 65.6	0.5709	0.5748
	6 M	123.3 ± 72.2	0.4693	123.7 ± 49.7	0.8754	0.668
	9 M	107.5 ± 32.8	0.3716	116.7 ± 41.0	0.5176	0.2436
BMI	BL	26.3 ± 4.8	-	24.8 ± 5.3	-	0.3673
(kg/m^2^)	1 M	26.4 ± 4.8	0.0957	24.8 ± 5.1	0.5817	0.3276
	3 M	26.3 ± 4.9	0.4172	24.8 ± 5.1	0.7188	0.3331
	6 M	26.2 ± 4.9	0.4331	24.5 ± 5.1	0.2953	0.3021
	9 M	26.2 ± 4.9	0.7971	24.5 ± 5.1	0.1322	0.3168

Values are expressed as mean ± SD. A comparison of HbA1c and inflammatory markers between the BL and each time point was performed using the Wilcoxon signed-rank test and was adjusted for multiple comparisons with a significance level of *: *p* < 0.005. Comparisons between groups were carried out using the Mann–Whitney *U* test with a significance level of *: *p* < 0.05.

**Table 5 antibiotics-11-01266-t005:** Correlations between changes in HbA1c, inflammatory markers, and clinical parameters between the FM-SRP group and the control group combined over nine months.

		Biomarker Measurements				Periodontal Clinical Measurements		
	HbA1c	hs-CRP	TNF-α	IL-6	MCP-1	PPD	BOP	PISA
Correlation *p* Value	(%)	(mg/dL)	(pg/mL)	(pg/mL)	(pg/mL)	(mm)	(%)	(mm^2^)
HbA1c (%)	-							

hs-CRP (mg/dL)	0.5683	-						
	<0.0001 *							
TNF-α (pg/mL)	0.4823	0.4681	-					
	0.0007 *	0.0010 *						
IL-6 (pg/mL)	0.2316	0.5416	0.5111	-				
	0.1215	0.0001 *	0.0003 *					
MCP-1 (pg/mL)	0.005	−0.068	0.0452	−0.0132	-			
	0.9737	0.6536	0.7657	0.9303				
PPD (mm)	0.5629	0.3157	0.3461	0.0507	0.0477	-		
	<0.0001 *	0.0326	0.0185	0.738	0.753			
BOP (%)	0.5061	0.3966	0.397	0.2429	−0.0072	0.7823	-	
	0.0003 *	0.0064	0.0063	0.1038	0.9624	<0.0001 *		
PISA (mm^2^)	0.5963	0.4738	0.4763	0.2608	−0.0163	0.8242	0.9353	-
	<0.0001 *	0.0009 *	0.0008 *	0.0801	0.9145	<0.0001 *	<0.0001 *	

Correlation coefficients and *p* values are shown in the table. Correlations between each parameter were evaluated using Spearman’s rank correlation coefficient. To assess the statistical significance of the correlations between the treatment and control groups, *p* values for multiple comparisons were adjusted using the Bonferroni correction, with a significance level of 0.05/28 tests *: *p* < 0.0017.

**Table 6 antibiotics-11-01266-t006:** Correlations between changes in HbA1c, inflammatory markers, and clinical parameters during nine months in the FM-SRP group.

		Biomarker Measurements				Periodontal Clinical Measurements		
	HbA1c	hs-CRP	TNF-α	IL-6	MCP-1	PPD	BOP	PISA
Correlation *p* Value	(%)	(mg/dL)	(pg/mL)	(pg/mL)	(pg/mL)	(mm)	(%)	(mm^2^)
HbA1c (%)	-							

hs-CRP (mg/dL)	0.5995	-						
	0.0020 *							
TNF-α (pg/mL)	0.6265	0.5997	-					
	0.0011 *	0.002						
IL-6 (pg/mL)	0.4632	0.5556	0.6806	-				
	0.0226	0.0048	0.0003 *					
MCP-1 (pg/mL)	−0.2718	−0.3011	−0.1872	−0.2157	-			
	0.1989	0.1527	0.3812	0.3114				
PPD (mm)	0.0967	0.2117	0.2647	−0.0476	−0.1439	-		
	0.6529	0.3208	0.2112	0.8252	0.5024			
BOP (%)	0.101	0.3296	0.2857	0.3287	−0.2106	0.0355	-	
	0.6386	0.1158	0.1759	0.1168	0.3232	0.8692		
PISA (mm^2^)	0.4163	0.613	0.5801	0.3777	−0.2454	0.2121	0.6809	-
	0.043	0.0014 *	0.0030 *	0.0688	0.2477	0.3197	0.0003 *	

Correlation coefficients and *p* values are shown in the table. Correlations between each parameter were evaluated using Spearman’s rank correlation coefficient. To assess the statistical significance of the correlations between the treatment and control groups, *p* values for multiple comparisons were adjusted using the Bonferroni correction, with a significance level of 0.05/28 tests *: *p* < 0.0017.

**Table 7 antibiotics-11-01266-t007:** Correlations between changes in HbA1c, inflammatory markers, and clinical parameters in the control group over nine months.

		Biomarker Measurements				Periodontal Clinical Measurements		
	HbA1c	hs-CRP	TNF-α	IL-6	MCP-1	PPD	BOP	PISA
Correlation *p* Value	(%)	(mg/dL)	(pg/mL)	(pg/mL)	(pg/mL)	(mm)	(%)	(mm^2^)
HbA1c (%)	-							

hs-CRP (mg/dL)	0.3187	-						
	0.1483							
TNF-α (pg/mL)	−0.0603	0.0543	-					
	0.79	0.8104						
IL-6 (pg/mL)	−0.0006	0.4608	0.3428	-				
	0.998	0.0309	0.1184					
MCP-1 (pg/mL)	0.1211	0.0781	0.4395	0.2158	-			
	0.5912	0.7299	0.0407	0.3348				
PPD (mm)	0.2595	−0.2422	−0.2454	−0.0434	0.1627	-		
	0.2435	0.2775	0.2711	0.8478	0.4694			
BOP (%)	0.0466	−0.0155	−0.1414	0.355	−0.0034	0.2917	-	
	0.837	0.9452	0.5303	0.1049	0.988	0.1878		
PISA (mm^2^)	0.0822	0.0028	−0.135	0.3774	−0.0164	0.4531	0.8257	-
	0.7161	0.99	0.5491	0.0834	0.9423	0.0342	<0.0001 *	

Correlation coefficients and *p* values are shown in the table. Correlations between each parameter were evaluated using Spearman’s rank correlation coefficient. To assess the statistical significance of the correlations between the treatment and control groups, *p* values for multiple comparisons were adjusted using the Bonferroni correction, with a significance level of 0.05/28 tests *: *p* < 0.0017.

## Data Availability

Data is contained within the article.

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
