# Peer review of "Effects of One-Stage Full-Mouth Scaling and Root Planing with Azithromycin on Diabetes and Periodontal Disease: A Randomized Controlled Trial"

_antibiotics, 2022, doi:10.3390/antibiotics11091266_

Round 1

Reviewer 1 Report

1. The title of the manuscript should be changed as: "Effects of one-stage full-mouth scaling and root planing with azithromycin on diabetes and periodontal disease: A randomized controlled trial".

2. The information about full-mouth scaling and root planing (FM-SRP) and azithromycin (AZM) should be included in the Introduction part.

3. Are all patients included in this study type 2 diabetes? This should be mentioned.

4. The bacterial names have to be italic.

5. Several typing errors should be corrected.

Author Response

Dear Reviewer 1

We would like to thank you and the reviewers for the valuable comments on our manuscript. We have taken all these comments into account and submit, herewith, a revised version of our paper. We have addressed all the comments by reviewers, as indicated in the attached pages, and we hope that the explanations are satisfactory.

We hope that the revised version of our paper is now suitable for publication in antibiotics, and we look forward to hearing from you at the earliest. Changes are indicated in red in the article.

  1. The title of the manuscript should be changed to: "Effects of one-stage full-mouth scaling and root planing with azithromycin on diabetes and periodontal disease: A randomized controlled trial".

Thank you for your advice. I changed the title according to your suggestion.

  1. The information about full-mouth scaling and root planing (FM-SRP) and azithromycin (AZM) should be included in the Introduction part.

Thank you for pointing out.

A description of FM-SRP and Azithromycin was added to the introduction part.

  1. Are all patients included in this study with type 2 diabetes? This should be mentioned.

Thank you for your suggestion.

Added to the introduction and materials and methods that all subjects were a type 2 diabetic

  1. The bacterial names have to be italic.

Corrected.

  1. Several typing errors should be corrected.

Fixed some typing errors.

Reviewer 2 Report

1. Please change “Methods and Materials” to be after Introduction, before Results

2. In 4.2 “Randomization and Allocation Concealment”, you have written “but the ethics committee has stated that it is ethically problematic to perform FM-SRP alone in diabetic patients who are susceptible to infections. As pointed out, the control group was only scaling…the control group was only scaling”. Do you mean you are against the ethics committee’s standpoint? Has the Ethics Committee approved what you have done on the control group for performing scaling only?

3. In 4.7 Statistical Analysis, you have written “A sample size calculation suggested that at least six patients were needed to demonstrate a 0.5% decrease in HbA1c after FM-SRP [95% power, α = 0.01, a standard deviation of 0.18 (JMP14.0.0)] [9]”. Please state clearly what parameters and formula you used to arrive the result of “6 patients”. Please clarify why your sample size was determined to be 6 but you end up recruiting more than 20 in each groups. Please also confirm [9] is the correct reference, as I cannot find [9] to support your statement. 

4. Please write a few sentences on description and appropriate reference on PISA in materials and methods

Author Response

Dear Reviewer 2

We would like to thank you and the reviewers for the valuable comments on our manuscript. We have taken all these comments into account and submit, herewith, a revised version of our paper. We have addressed all the comments by reviewers, as indicated in the attached pages, and we hope that the explanations are satisfactory.

We hope that the revised version of our paper is now suitable for publication in antibiotics, and we look forward to hearing from you at the earliest. Changes are indicated in red in the article.

  1. Please change “Methods and Materials” to be after Introduction, before Results

Thank you for your advice.

We moved “Methods and Materials” to after Introduction, before Results.

  1. In 4.2 “Randomization and Allocation Concealment”, you have written “but the ethics committee has stated that it is ethically problematic to perform FM-SRP alone in diabetic patients who are susceptible to infections. As pointed out, the control group was only scaling…the control group was only scaling”. Do you mean you are against the ethics committee’s standpoint? Has the Ethics Committee approved what you have done on the control group for performing scaling only?

Sorry for the unclear description.

The Ethics Committee pointed out that FM-SRP without antibiotics for diabetic patients, who are susceptible to infection, is ethically problematic due to the infection problem. In other words, it is pointed out that FM-SRP under AZM administration is good, but there is a problem with FM-SRP without the administration of antibiotics. Therefore, the control group was only performed as scaling, which is a low-risk procedure. SRP and scaling are different procedures.

  1. In 4.7 Statistical Analysis, you have written “A sample size calculation suggested that at least six patients were needed to demonstrate a 0.5% decrease in HbA1c after FM-SRP [95% power, α = 0.01, a standard deviation of 0.18 (JMP14.0.0)] [9]”. Please state clearly what parameters and formula you used to arrive at the result of “6 patients”. Please clarify why your sample size was determined to be 6 but you end up recruiting more than 20 in each group. Please also confirm [9] is the correct reference, as I cannot find [9] to support your statement. 

The sample size was determined using the formula (Barker, C.(2011). Power and Sample Size Calculations in JMP. Technical Report, Cary NC: SAS Institute Inc.) in the statistical software (JMP14.0.0).

Details of the calculation formula are provided at the following URL.

https://www.jmp.com/content/dam/jmp/documents/en/technical-reports/powerAndSampleSize.pdf.

The reference site is 2.2. "Two Samples: Comparing Means."

Calculations were made using the standard deviations of HbA1c for the intervention and non-intervention groups from reference [9].

A table of standard deviations used to calculate the sample size is shown below.

Table is sent by attached file. 

In this study, the calculated sample size of 6 people is the minimum value, and considering dropouts, etc., we set the sample size to about 20 people who can be recruited.

  1. Please write a few sentences on description and appropriate reference on PISA in materials and methods

Thank you for your advice.

We added a few sentences on description and appropriate references on PISA in materials and methods.
